# VMoBA: Mixture-of-Block Attention for Video Diffusion Models

**Jianzong Wu**[1,2]*, **Liang Hou**[2], **Haotian Yang**[2], **Ye Tian**[1]
**Pengfei Wan**[2], **Di Zhang**[2], **Yunhai Tong**[1]
[1]Peking University [2]Kling Team, Kuaishou Technology
**Code:** https://github.com/KwaiVGI/VMoBA
*Email: jzwu@stu.pku.edu.cn*

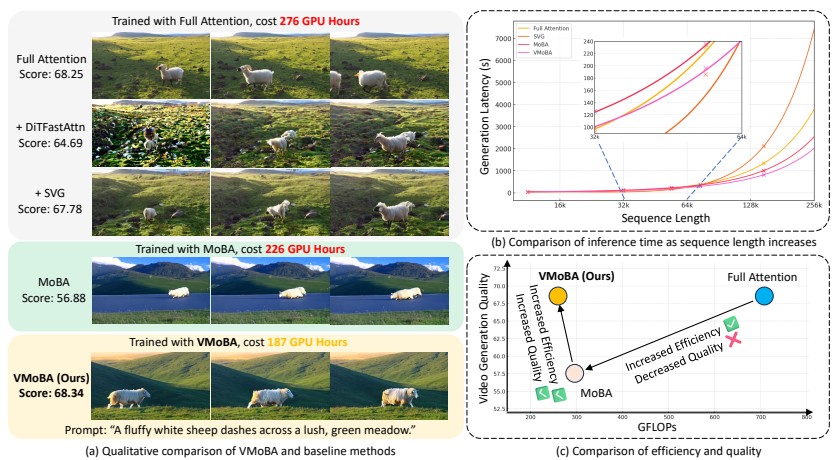

Figure 1: (a) VMoBA performs better than Full Attention while reducing training time. (b) VMoBA is faster when the sequence length goes higher. The 'x' points represent tested latencies, and the lines are fitted with quadratic functions. (c) VMoBA is an efficient and effective adaptation of MoBA.

## Abstract

The quadratic complexity of full attention mechanisms poses a significant bottleneck for Video Diffusion Models (VDMs) aiming to generate long-duration, high-resolution videos. While various sparse attention methods have been proposed, many are designed as training-free inference accelerators or do not optimally capture the unique spatio-temporal characteristics inherent in video data when trained natively. This paper introduces Video Mixture of Block Attention (VMoBA), a novel sparse attention mechanism specifically adapted for VDMs. Motivated by an in-depth analysis of attention patterns within pre-trained video transformers, which revealed strong spatio-temporal locality, varying query importance, and head-specific concentration levels, VMoBA enhances the original MoBA framework with three key modifications: (1) a layer-wise recurrent block partition scheme (1D-2D-3D) to dynamically adapt to diverse spatio-temporal attention patterns and improve efficiency; (2) global block selection to prioritize the most salient query-key block interactions across an entire attention head; and (3) threshold-based block selection to dynamically determine the number of attended blocks based on their cumulative similarity. Extensive experiments demonstrate that VMoBA significantly accelerates the training of VDMs on longer sequences, achieving 2.92× FLOPs and 1.48× latency speedup, while attaining comparable or even superior generation quality to full attention. Furthermore, VMoBA exhibits competitive performance in training-free inference, offering 2.40× FLOPs and 1.35× latency speedup for high-res video generation.

---

*The work is done during Jianzong Wu's internship in Kling Team, Kuaishou Technology.

# 1 INTRODUCTION

Modeling long-sequence data is a critical challenge in modern deep learning, particularly for Video Diffusion Models (VDMs) for generating high-resolution, long-duration videos (Saharia et al., 2022; Rombach et al., 2022; Guo et al., 2024; Wang et al., 2023b;a; Chen et al., 2023; 2024a; Ma et al., 2024b; Brooks et al., 2024; 2023; Kong et al., 2024; Wang et al., 2025). The ubiquitous full attention mechanism, while powerful, exhibits a computational complexity that scales quadratically with the sequence length (Vaswani et al., 2017). This bottlenecks the ability to process longer sequences. For example, a 720p video can exceed 76K tokens (Section 4).

To mitigate this, various sparse attention mechanisms have been proposed (Lu et al., 2025; Yuan et al., 2025; Ma et al., 2024a; Yuan et al., 2024; Zhang et al., 2025b; Xi et al., 2025; Zhang et al., 2025a). The core idea is to let each query interact with only a subset of key-value pairs, thereby reducing computational complexity. Sparse attentions for VDMs are primarily designed as training-free methods (Xi et al., 2025; Yuan et al., 2024; Zhang et al., 2025a). However, these methods show suboptimal results without training (Section 4). Leveraging sparse attention to accelerate the **training** of VDMs, while being a realistic demand, remains a relatively underexplored domain.

Recently, Mixture of Block Attention (MoBA) (Lu et al., 2025) is introduced as a sparse attention mechanism for training. It demonstrates effectiveness in Large Language Models (LLMs) training for long-context data. Unfortunately, our initial attempt to directly apply MoBA to the training of VDMs yields unsatisfactory results, as illustrated in Fig. 1a, where MoBA shows a significant quality drop (VBench Score 68.25 → 56.88). We attribute this to MoBA's design being primarily optimized for textual data, where the locality spans on the 1D space, leading to a performance gap for the video generation task, where the locality lays on the 3D space.

To bridge this gap, we conduct a task-specific analysis of attention mechanisms within pre-trained Video Diffusion Transformers (DiTs). Our analysis (detailed in Section 3) reveals several key insights:

***Observation 1: Full attention map has 1-2-3D attention patterns.*** Video data exhibits strong spatio-temporal locality, with attention mechanisms showing focusing on tokens within local one, two, or three dimensional neighborhoods (Fig. 3). MoBA's original key block partitioning – flattening the latent space into a one-dimensional sequence and then uniformly splitting into blocks – fundamentally disrupts this locality. Spatially adjacent tokens may be assigned to different blocks, diluting the representational power of block means. Existing sparse attention methods, such as SparseVideoGen (Xi et al., 2025), categorizes attention heads into spatial and temporal. However, the role of any particular head can shift with changes in prompt, diffusion step, or layer. Lacking a reliable prior on which heads should be spatial or temporal, SparseVideoGen computes a small *pilot* attention before each layer to classify head types, incurring substantial overhead that, as sequence length grows, can render it even slower than full attention, as shown in Fig. 1b.

***Observation 2: Queries have various importance.*** Attention maps indicate that different query tokens receive varying degrees of attention, and their top similarity scores with key tokens can differ significantly (Fig. 4). Selecting the same number of key blocks for each query might under-allocate resources to queries that inherently have stronger affinities with keys.

***Observation 3: Heads have different concentration levels.*** The concentration level of similarity scores varies across different attention heads (Fig. 5). Some heads exhibit highly concentrated similarities, while others show more smoothed patterns. A fixed Top-K selection, as in MoBA, may not be optimal because it ignores that different heads have different similarity-score distributions.

Motivated by these observations, we propose **VMoBA**, a sparse attention specifically adapted for VDMs training. There are three key innovations corresponding to the observations:

***Innovation 1: 1-2-3D block partition.*** Key blocks are partitioned using a cyclical 1D-2D-3D scheme across layers. This allows the model to learn appropriate spatio-temporal attention patterns during training. It also improves efficiency compared to a uniform 3D partitioning.

***Innovation 2: Query-global block selection.*** For each attention head, the top-scoring key blocks are selected from a global pool aggregated across all query-key block products. This prioritizes blocks corresponding to the most *important* query-key interactions.

*Innovation 3: Threshold-based block selection.* We dynamically determine the number of selected blocks based on their cumulative similarity score, rather than a fixed Top-K. This allows for a more adaptive approximation of full attention by catering to the diverse concentration levels across heads.

Through these targeted innovations, VMoBA enables efficient training of VDMs on longer sequences, significantly outperforming vanilla MoBA and achieving comparable or superior quality to full attention, as shown in Fig. 1a and Fig. 1c. Furthermore, VMoBA demonstrates competitive performance in training-free settings, offering substantial speedups for long sequences, as shown in Fig. 1b.

Our contributions are summarized as follows: **1)** We conduct an in-depth analysis of attention mechanisms in video DiTs, revealing key observations, including spatio-temporal characteristics, varying query importance, and diverse similarity distribution patterns. **2)** We propose VMoBA, the first mixture-of-block sparse attention specifically for training VDMs on long sequences. VMoBA incorporates novel layer-wise recurrent block partition, global block selection, and threshold-based block selection strategies. **3)** Extensive experiments demonstrate that VMoBA significantly accelerates training and inference for long video sequences while maintaining or improving generation quality compared to full attention and baseline sparse attention methods.

## 2 RELATED WORK

**Diffusion model inference acceleration.** The technologies of inference acceleration broadly fall into two categories. First, methods that reduce the number of denoising steps. These approaches modify only the inference procedure. Many edit the samplers or noise schedules to cut down diffusion timesteps (Song et al., 2020; Liu et al., 2022; Lu et al., 2022; Liu et al., 2023; Song et al., 2023; Luo et al., 2023). For example, high-order ODE solvers and multi-step methods can generate videos in tens of steps instead of hundreds (Liu et al., 2022; Lu et al., 2022). Other works exploit the redundancy between successive diffusion steps: by caching and reusing features (e.g., attention maps), they avoid repeated computation (Ma et al., 2023; Chen et al., 2024b; Zhao et al., 2024; Lv et al., 2024; Liu et al., 2025). However, these training-free methods are highly sensitive to hyperparameters and can produce degraded fidelity (Zhao et al., 2024; Chen et al., 2024b; Liu et al., 2025). Second, model distillation techniques train a student network to mimic the original model with fewer steps (Salimans & Ho, 2022; Yin et al., 2024). However, Distillation needs extra training on large-scale datasets, and the student model often generates on the same or smaller resolutions, struggling to scale the data to larger resolutions (Salimans & Ho, 2022).

**Efficient attention mechanisms.** Linear attention methods, such as state-space models (Gu & Dao, 2023; Dao & Gu, 2024; Liu et al., 2024; Wang et al., 2024a) and RNN-derived architectures (Peng et al., 2023) achieve linear complexity by reformulating the attention operation. However, these schemes typically cannot be seamlessly interchanged with full attention. Moreover, they are observed falling short on some tasks (Yu & Wang, 2024) compared with full attention. Alternatively, sparse attention mechanisms limit each token's attended set. Approaches like MoBA (Lu et al., 2025), NSA (Yuan et al., 2025), and Block-Attention (Ma et al., 2024a) allow Transformers to transition between full and sparse attention modes seamlessly. For video diffusion models, specialized sparse attention techniques have been introduced to accelerate generation at inference time, including Sliding Tile Attention (Zhang et al., 2025b), VSA (Zhang et al., 2025c), DiTFastAttn (Yuan et al., 2024), SparseVideoGen (Xi et al., 2025), and SpargeAttn (Zhang et al., 2025a). However, these techniques often serve training freely, offering limited benefit for speeding up model training. On the contrary, our proposed VMoBA is a sparse attention mechanism for video diffusion models. It accelerates inference with lossless quality and speeds up model training towards longer sequences.

## 3 METHOD

### 3.1 OVERVIEW

VMoBA operates in three steps, as illustrated in Fig. 2. **Step 1: Partition and Mean Key Blocks.** The input key tensor $\mathbf{K}$ is first partitioned into non-overlapping blocks. Crucially, the partitioning strategy (1D, 2D, or 3D) is determined layer-wise, as detailed in Section 3.2. After partitioning, the mean of each key block is computed, resulting in a set of key blocks $\mathbf{B}$. **Step 2: Select Key Blocks.** For

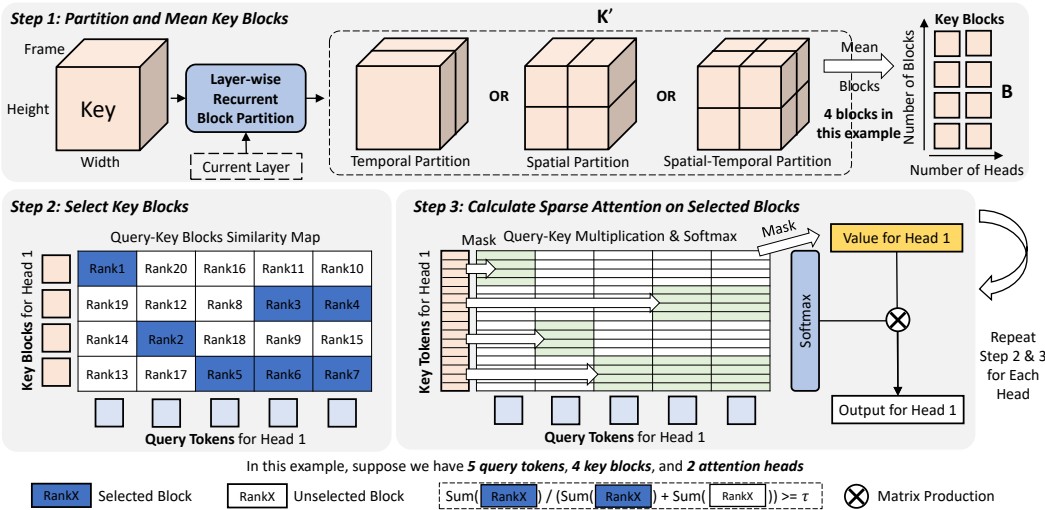

Figure 2: **The overall pipeline of VMoBA.** We first partition key blocks with Layer-wise Recurrent BLock Partition, then select the blocks using Global Block Selection and Threshold-based Block Selection. Finally, the attention is computed only with the selected blocks.

each attention head, a similarity map is computed between all query tokens **Q** and all the key blocks derived in Step 1. Instead of selecting a fixed number of key blocks for each query independently, VMOBA employs two essential modifications: 1) Global Block Selection (Section 3.3): The most significant query-key block interactions are identified from a global pool across all queries for the current head. 2) Threshold-based Selection (Section 3.4): The number of selected key blocks is dynamically determined based on their cumulative similarity score relative to the total similarity, controlled by a threshold $\tau$. **Step 3: Calculate Sparse Attention on Selected Blocks.** Each query token performs attention only with the key-value pairs belonging to its selected blocks. The outputs from all attention heads are concatenated to form the final attention layer output. Note that while this describes the conceptual process, our implementation leverages FlashAttention (Dao et al., 2022) for efficient training, ensuring the output is equivalent to this theoretical procedure.

**Computational complexity analysis (FLOPs).** The computational complexity of VMOBA primarily consists of two parts: 1) Block Selection: Calculating the similarity matrix between $s$ query tokens (each of dimension $d$) and $N_b$ key blocks. This is approximately $O(sdN_b) = O(s^2d/s_b)$, where $s_b$ is block size. 2) Sparse Attention: Performing attention between $s$ query tokens and their selected $k_{\text{avg}}$ key-value pairs on average. The complexity is roughly $O(sk_{\text{avg}}s_bd)$. Note that here we omit the attention head, assuming $d$ is the whole hidden dimension with all the heads. The total FLOPs can be approximated as $O(sd(s/s_b + k_{\text{avg}}s_b))$. This indicates that larger block sizes and fewer selected blocks per query contribute to lower FLOPs.

## 3.2 LAYER-WISE RECURRENT BLOCK PARTITION

Video data inherently possesses strong spatio-temporal locality. Our analysis of attention maps from a full attention DiT, Wan 2.1 1.3B (Wang et al., 2025), is illustrated in Fig. 3. We find several representative attention patterns across layers. Specifically, different layers might focus on temporal, spatial, or combined spatio-temporal neighborhoods. For example, layer 27 prioritizes interactions along the temporal axis (1D neighbor), layer 3 focuses on spatial relationships within a frame (2D neighbor), and layer 20 attends to local 3D spatio-temporal volumes (3D neighbor). Specifically, for the 3D neighbor pattern, the

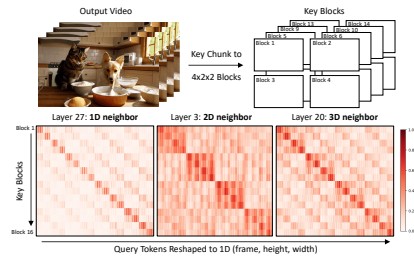

Figure 3: The query-key block attention map shows 1-2-3D distinct patterns.

queries belonging to the first block attend primarily to the 1st, 2nd, 3rd, and 5th key blocks, which are accurate themselves and their neighbors in width, height, and time.

To accommodate these varying patterns without costly dynamic selection (Xi et al., 2025), VMoBA employs a **Layer-wise Recurrent Block Partition** strategy. We define three types of key block partitioning: 1) Temporal (1D) Partition. Blocks are formed along the time axis, grouping tokens from the near frames. 2) Spatial (2D) Partition. Blocks are formed along the height and width dimensions, grouping tokens from the local spatial position across all frames. 3) Spatio-temporal (3D) Partition. Blocks are formed as local 3D patches in the spatio-temporal latent space. The three partition functions are performed recursively across layers in our model, which can be formulated as:

$$\mathbf{K}' = \text{rearrange}(\mathbf{K} \in (T\ H\ W)) \rightarrow \begin{cases} (N_{b_1}^T)\ (s_{b_1}^T\ H\ W), & \text{if } l \bmod 3 = 0 \\ (N_{b_2}^H\ N_{b_2}^W)\ (T\ s_{b_2}^H\ s_{b_2}^W), & \text{if } l \bmod 3 = 1 \\ (N_{b_3}^T\ N_{b_3}^H\ N_{b_3}^W)\ (s_{b_3}^T\ s_{b_3}^H\ s_{b_3}^W), & \text{if } l \bmod 3 = 2 \end{cases} \quad (1)$$

$$\mathbf{B} = \text{mean}(\mathbf{K}') \in \begin{cases} (N_{b_1}^T), & \text{if } l \bmod 3 = 0 \\ (N_{b_2}^H\ N_{b_2}^W), & \text{if } l \bmod 3 = 1 \\ (N_{b_3}^T\ N_{b_3}^H\ N_{b_3}^W), & \text{if } l \bmod 3 = 2 \end{cases} \quad (2)$$

where $\mathbf{K}'$ is the key after rearrange, and $\mathbf{B}$ is the key block tensor. $l$ is the layer number. $T$, $H$, and $W$ represent the frame, height, and width of the video latent. $N_{b_x}^T$, $N_{b_x}^H$, and $N_{b_x}^W$ are the number of blocks in time, height, and width dimensions for each partition pattern, respectively. $s_{b_x}^T$, $s_{b_x}^H$, and $s_{b_x}^W$ are the corresponding block sizes. The subscript $x \in \{1, 2, 3\}$ indicates the partition pattern (1-2-3D). We omit batch size and hidden size here for simplicity. Thanks to the training nature of VMoBA, this cyclical 1D-2D-3D scheme allows the model to learn spatio-temporal attention patterns automatically. This approach improves efficiency by reducing the total number of blocks compared to a uniform 3D partition across all layers, as VMoBA's speed is sensitive to the total block number. Furthermore, this structured partitioning helps ensure that semantically related tokens are more likely to be grouped within the same block, leading to more representative mean key blocks.

## 3.3 GLOBAL BLOCK SELECTION

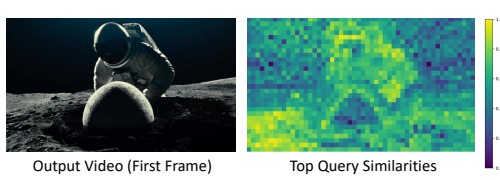

Output Video (First Frame)  Top Query Similarities

Figure 4: Summation of top 25% query-key similarities. Different queries have varying importance.

Full attention maps indicate that different query tokens receive varying degrees of attention, and their top similarity scores with key tokens can differ significantly. As shown in Fig. 4, the sum of top 25% query-key similarities varies greatly across queries. For example, regions in "astronaut" and "ground" have high top similarity scores while regions in "ball" and "sky" have lower scores. This indicates that some queries inherently have stronger affinities or *importance* regarding key interactions. MoBA's original strategy selects a fixed number of key blocks for **each query independently**. This can under-allocate resources to queries that inherently have stronger affinities with keys and could benefit from interacting with more blocks, or over-allocate to less important queries. To address this, the proposed VMoBA implements **Global Block Selection** algorithm. For each attention head, instead of selecting blocks on a per-query basis, we first compute all query-key block similarities. Then, the top-scoring key blocks are selected from this global pool of similarities, aggregated across all query-key block products. This prioritizes blocks corresponding to the most important overall query-key interactions, allowing sparsity to be allocated more effectively based on the collective signal strength of interactions. This can be formulated as:

$$\mathbf{M}_i = \text{TopkMask}(\mathbf{q}_i \mathbf{b}_i^T,\ k), \quad i \in \{1, 2, ..., h\} \quad (3)$$

where the $\text{TopkMask}(a, b)$ function selects the top $b$ values in $a$ and returns a mask with selected positions set to 1. $i$ is the attention head index, from 1 to $h$, the number of attention heads. $\mathbf{q}_i$ in shape $s \times d$] contains all the query tokens for head $i$, and $\mathbf{b}_i$ in shape $N_b \times d$ is all the key blocks for head $i$.

$\mathbf{q}_i\mathbf{b}_i^T$ is a matrix production to get their similarity matrix. $\mathbf{M}_i$ in shape $N_b \times s$ is the selected query-key block pair mask. From this formula, VMoBA selects blocks globally, considering the different importance of queries. Here $k$ is the number of selected blocks, further defined in Section 3.4.

## 3.4 THRESHOLD-BASED BLOCK SELECTION

The concentration level of similarity scores varies not only across queries but also across different attention heads. Some heads exhibit highly concentrated similarities towards a few query-key block pairs, while others show more diffuse or smoothed patterns. This is illustrated in Fig. 5, which shows the distribution of sorted query-key block similarities for different heads. The varying slopes indicate different concentration levels. Specifically, head 4 rapidly increases the similarity on the most right part (the largest similarities), while head 1 requires more blocks to capture a significant portion of the total similarity. For a better understanding of the concentration level, we draw the 30% and 50% cutoff lines for each head, observing significant differences across the heads. The number of query-block pairs required to reach 50% cumu-

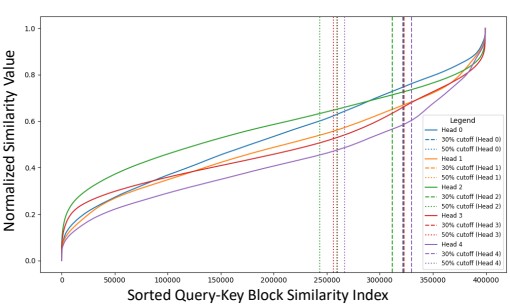

Figure 5: Sorted query-key block similarity and top 30%/50% cutoff lines. The right parts of the cutoff lines contain corresponding cumulative summations of query-key block similarity.

lative similarity for heads 1 and 4 differs by nearly 25,000, which is highly significant. Under such an observation, a fixed Top-K selection of blocks, as in the original MoBA, which ignores these head-specific distribution differences, may not be optimal. It might select too few blocks for diffuse heads, thus losing information, or too many low-similarity blocks for concentrated heads, thus wasting computation. Therefore, VMoBA introduces **Threshold-based Block Selection** to determine the number of selected blocks dynamically. After obtaining the global query-key block similarity scores, we sort these similarity scores in descending order. We then select blocks starting from the highest similarity, accumulating their normalized similarity scores until this cumulative sum exceeds a predefined threshold. The process to dynamically define $k$ can be written as:

$$k = \min\Big\{k' \mid \sum_{j=1}^{k'} \text{Sorted}(\hat{S}_j) \geq \tau\Big\}, \tag{4}$$

where $\hat{S} = \mathbf{q}_i\mathbf{b}_i^T$ is the computed query-key block similarity matrix. $\tau$ is a hyperparameter to define the cumulative similarity threshold. This adaptive approach allows VMoBA to approximate full attention more effectively by catering to the diverse concentration levels across heads. It ensures that high-similarity pairs are captured while efficiently pruning low-similarity ones, acting as an effective *compression* of sparsity based on informational content, rather than a fixed $k$.

## 4 EXPERIMENTS

### 4.1 SETUP

**Datasets.** For training VMoBA and the baseline models, we use the Koala-36M dataset (Wang et al., 2024b). We train at different resolutions to assess VMoBA's generalization capability. Importantly, in each experimental setting, all models are trained on the identical data to eliminate any confounding effects due to data variation. To evaluate the video generation ability, we use the VBench prompt list (Huang et al., 2024). For training-free inference experiments, we use the original prompt list. For the training-based results, we use the prompts after optimization (Yang et al., 2024) because training prompts from Koala-36M (Wang et al., 2024b) are long.

**Metrics.** We adopt five assessment aspects of VBench (Huang et al., 2024) to evaluate the video generation ability. They are Overall Consistency (TextConsis), Dynamic Degree (Dynamic), Background

Table 1: **Training-free inference comparison** of VMoBA and baseline methods.

| Video Size | Method | Sparsity | Similarity | Quality | | | | Efficiency | |
|---|---|---|---|---|---|---|---|---|---|
| | | | PSNR ↑ | TextConsis ↑ | BGConsis ↑ | ImageQual ↑ | SubConsist ↑↓ | FLOPs ↓ | Latency ↓ |
| | FullAttn | - | - | 27.85% | 94.47% | 62.06% | 90.70% | 282.64T | 103s |
| $81 \times 480 \times 832$ | DiTFastAttn (Yuan et al., 2024) | 0.50 | 22.67 | 27.34% | 91.68% | 60.27% | 90.57% | 182.73T (1.54×) | 89s (1.18×) |
| | SVG (Xi et al., 2025) | 0.50 | 12.64 | 27.30% | 93.67% | 60.98% | 89.84% | 182.92T (1.54×) | 90s (1.17×) |
| 33K Tokens | MoBA (Lu et al., 2025) | 0.25 | 15.54 | 28.47% | 94.84% | 60.06% | 89.90% | 133.50T (2.12×) | 126s (0.83×) |
| | VMoBA | 0.31 | 16.00 | 27.62% | 93.74% | 60.05% | 89.65% | 149.00T (1.90×) | 104s (1.01×) |
| | FullAttn | - | - | 27.99% | 93.74% | 63.87% | 92.56% | 1246.78T | 406s |
| $81 \times 720 \times 1280$ | DiTFastAttn (Yuan et al., 2024) | 0.50 | 24.50 | 26.09% | 92.03% | 65.59% | 92.25% | 720.06T (1.73×) | 310s (1.31×) |
| | SVG (Xi et al., 2025) | 0.50 | 25.85 | 27.64% | 92.11% | 63.56% | 93.98% | 720.50T (1.73×) | 314s (1.29×) |
| 76K Tokens | MoBA (Lu et al., 2025) | 0.25 | 20.46 | 28.47% | 91.46% | 64.16% | 91.42% | 457.20T (2.73×) | 360s (1.13×) |
| | VMoBA | 0.31 | 18.80 | 28.06% | 92.85% | 64.39% | 92.08% | 519.75T (2.40×) | 300s (1.35×) |

Table 2: **Training-based results.** Training time is measured in GPU hours. Training-free methods apply to tuned Full Attention models.

| Video Size | Method | Sparsity | Quality | | | | | Efficiency | |
|---|---|---|---|---|---|---|---|---|---|
| | | | TextConsis ↑ | Dynamic ↑ | BGConsis ↑ | ImageQual ↑ | SubConsist ↑ | FLOPs ↓ | Training Time ↓ |
| | Pretrained Wan 2.1 | - | 28.22% | 62.97% | 93.94% | 63.81% | 92.40% | 705.02T | - |
| $93 \times 576 \times 1024$ | FullAttn | - | 24.61% | 61.58% | 94.69% | 69.49% | 90.86% | 705.02T (1.00×) | 276h (1.00×) |
| | + DiTFastAttn (Yuan et al., 2024) | 0.50 | 22.60% | 61.57% | 88.59% | 67.36% | 83.33% | 423.22T (1.66×) | - |
| | + SVG (Xi et al., 2025) | 0.50 | 23.37% | 62.43% | 93.84% | 68.89% | 90.38% | 423.55T (1.66×) | - |
| 55K Tokens | MoBA (Lu et al., 2025) | 0.25 | 23.06% | 5.80% | 97.60% | 63.73% | 94.30% | 282.69T (2.49×) | 226h (1.22×) |
| | VMoBA | 0.19 | 25.88% | 56.91% | 96.76% | 67.45% | 94.72% | 248.68T (2.83×) | 187h (1.48×) |
| | Pretrained Wan 2.1 | - | 27.11% | 56.58% | 92.32% | 63.49% | 92.66% | 724.97T | - |
| $141 \times 480 \times 832$ | FullAttn | - | 23.92% | 43.01% | 92.30% | 64.36% | 92.58% | 724.97T (1.00×) | 262h (1.00×) |
| | + DiTFastAttn (Yuan et al., 2024) | 0.50 | 21.39% | 44.35% | 89.47% | 63.64% | 86.92% | 434.31T (1.67×) | - |
| | + SVG (Xi et al., 2025) | 0.50 | 21.37% | 43.67% | 93.72% | 63.72% | 92.38% | 434.64T (1.67×) | - |
| 56K Tokens | MoBA (Lu et al., 2025) | 0.25 | 23.22% | 11.97% | 95.39% | 65.07% | 93.40% | 289.16T (2.51×) | 209h (1.25×) |
| | VMoBA | 0.18 | 23.71% | 31.36% | 93.06% | 67.66% | 93.78% | 248.39T (2.92×) | 182h (1.44×) |

Consistency (BGConsis), Imaging Quality (ImageQual), and Subject Consistency (SubConsist). Furthermore, we use PSNR (Fardo et al., 2016) to evaluate the similarity of sparse and full attention generated videos in the training-free inference setting. We also count the generation latency, training time, and the corresponding speedup to compare the efficiency of methods.

**Baselines.** We compare VMoBA with its most relevant native training-targeted sparse attention method, MoBA (Lu et al., 2025). We also compare with two training-free sparse attention methods designed for DiTs, DiTFastAttn (Yuan et al., 2024) and SVG (Xi et al., 2025). Full Attention (Wang et al., 2025) is considered a ground-truth in the training-free setting. In the training-based experiments, it becomes a comparable baseline method. For a thorough insight into the results, we only compare the **Attention** mechanism with baselines, eliminating other tricks like cache or quantization.

**Implementation details.** We use Wan 2.1 1.3B (Wang et al., 2025) as the base model for all the experiments. For VMoBA, we set the threshold $\tau$ to 0.25, while its token sparsity may vary across different settings (always smaller than $\tau$ because of the selection algorithm). The block size and block number differ in various resolutions. We primarily set the number of spatial-temporal blocks to 72, which is sufficient for most cases. Following previous works (Xi et al., 2025; Zhao et al., 2024; Li et al., 2024; Lv et al., 2024; Liu et al., 2025), we remain the first 25% of denoising steps with full attention. Training experiments are finished with 2000 steps. Additional implementation details can be found in the Appendix.

## 4.2 QUANTITATIVE RESULTS

**Training-free inference.** We conduct training-free inference experiments on two resolutions. One is the original resolution of Wan (Wang et al., 2025) ($81 \times 480 \times 832$, 33K tokens), the other is an extended token length ($81 \times 720 \times 1280$, 76K tokens). The results are presented in Table 1. At the original resolution, VMoBA achieves a 1.01× speedup over Full Attention, surpassing MoBA's 0.83×. While its speedup is slightly less than training-free methods, VMoBA demonstrates strong generation quality. It ranks second in TextConsis and BGConsis, with other scores being comparable (within 1% of the top baseline). Regarding similarity to Full Attention, as measured by PSNR, VMoBA is second only to DiTFastAttn. When extending to longer sequences, VMoBA's advantages in speed and quality become more pronounced. Specifically, VMoBA achieves a 1.35× speedup, which is 19% higher than vanilla MoBA. This improvement is primarily because MoBA's 1D partitioning leads to excessive number of blocks. In terms of quality, VMoBA leads in BGConsis, is second in TextConsis and ImageQual, and is comparable in SubConsist. Its PSNR is lower than some methods,

indicating that VMoBA's generated videos have less similarity to Full Attention outputs in this longer token setting. This might reflect a characteristic of VMoBA: for long tokens, it can generate videos that are not identical to Full Attention but still maintain high quality. In summary, despite being designed for training, VMoBA demonstrates acceptable performance in training-free settings.

**Training on longer sequences.** To evaluate VMoBA's ability to fine-tune pre-trained models on longer sequences, we train Wan (Wang et al., 2025) on two extended resolutions: 1) Spatially extended video: 93x576x1024 (55K tokens). 2) Temporally extended video: 141x480x832 (56K tokens). Results are shown in Table 2. For spatially extended videos, VMoBA, with the lowest token sparsity (0.19) and training time, achieves quality comparable to or even exceeding Full Attention. The VBench mean score is 68.34 for VMoBA and 68.25 for Full Attention. Full Attention required 276 GPU Hours for training, while VMoBA only needed 187 GPU Hours, resulting in a $1.48\times$ speedup. The vanilla MoBA's score is particularly low on Dynamic Degree (5.80%), indicating its outputs have minimal dynamics and are almost static. This is likely because MoBA's 1D block partitioning struggles to capture spatio-temporal motion information effectively. DiTFastAttn[†] and SVG[†], applied to the tuned Full Attention model, yield lower scores than Full Attention and do not offer training acceleration. For temporally extended videos, VMoBA achieves a $1.44\times$ speedup over Full Attention, again delivering comparable or superior performance. Notably, VMoBA outperforms Full Attention by 3.3% in Image Quality. In summary, the proposed VMoBA accelerates DiT training significantly while ensuring comparable or improved generation quality on longer sequences.

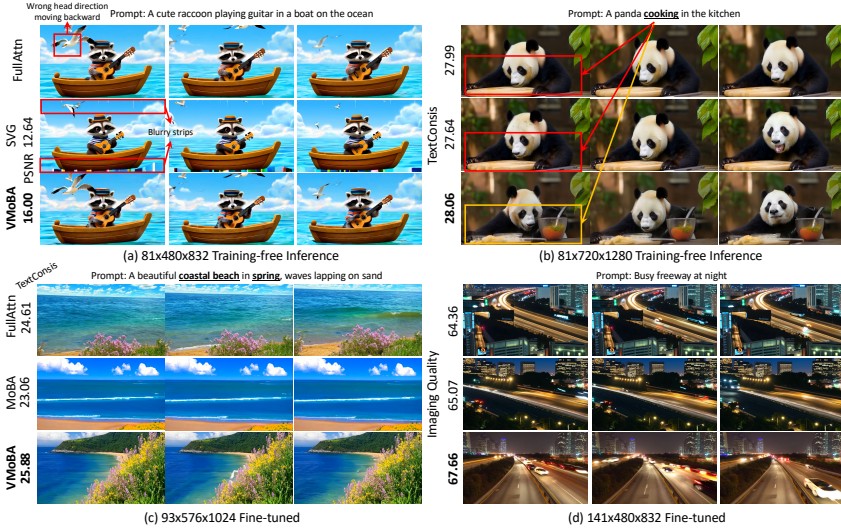

Figure 6: **Qualitative comparison** of VMoBA and baseline methods.

## 4.3 QUALITATIVE RESULTS

Fig. 6 showcases four examples, illustrating qualitative results. (a) Displays training-free inference results at the original resolution. Even with a sparsity of 0.5, SVG (Xi et al., 2025) exhibits clear blurry strips at the top and bottom of the video, contributing to its very low PSNR (12.64). In contrast, VMoBA closely reproduces the video content. Interestingly, in the ground truth video generated by Full Attention, the seagull's head appears in an incorrect orientation, an artifact not present in VMoBA's generation. (b) Presents results for high-definition video ($81\times720\times1280$) with the prompt "A panda cooking in the kitchen." Full Attention generates a panda and a chopping board but misses the "cooking" aspect. VMoBA, however, generates dishes and condiments, aligning more closely with the prompt. This observation helps explain VMoBA's higher TextConsis score 28.06% vs. 27.99% for Full Attention. SVG's output is more similar to Full Attention's, resulting in a higher PSNR but a TextConsis score 0.42% lower than VMoBA's. (c) Illustrates results after training models on the spatially extended resolution. VMoBA generates an aesthetically pleasing scene of coastal waves in spring, consistent with the prompt. Full Attention, on the other hand, lacks a clear depiction of the coast. MoBA's output is notably static and less aesthetically appealing. This demonstrates that VMoBA, after training, can generate videos comparable to or better than Full Attention while

achieving a significant training speedup (1.48×). (d) shows results after training on a temporally extended resolution. VMoBA's output exhibits better image quality and more realistic scene depiction than other methods. In summary, VMoBA offers significant speed improvements for long sequences. After training, it can achieve visual quality consistent with, and in some aspects, even better than, full attention mechanisms, particularly in terms of prompt alignment and imaging quality.

Table 3: **Ablation studies.** 'DD', 'IQ', and 'SC' represent Dynamic, ImageQual, and SubConsis. Training time is measured in GPU hours.

(a) Key block partition method ablation.

| Method | DD ↑ | IQ ↑ | SC ↑ | Time ↓ |
|---|---|---|---|---|
| 1-2D | 55.49% | 58.51% | 86.12% | 176 |
| 1-3D | 28.57% | 66.71% | 91.34% | 187 |
| 2-3D | **57.01%** | 66.02% | **94.75%** | 202 |
| **1-2-3D** | 56.91% | **67.45%** | 94.72% | 187 |

(b) Block choice design ablation.

| Method | DD ↑ | IQ ↑ | SC ↑ |
|---|---|---|---|
| topk + local | 54.87% | 65.59% | 91.64% |
| threshold + local | 55.19% | 65.31% | 92.43% |
| topk + global | 55.29% | 64.58% | 92.86% |
| **threshold + global** | **56.91%** | **67.45%** | **94.72%** |

(c) Ablation on threshold $\tau$.

| $\tau$ | Sparsity | DD ↑ | IQ ↑ | SC ↑ | Time ↓ |
|---|---|---|---|---|---|
| 0.15 | 0.13 | 55.46% | 66.82% | 91.12% | 162 |
| **0.25** | 0.19 | **56.91%** | 67.45% | **94.72%** | 187 |
| 0.35 | 0.27 | 55.00% | 67.63% | 94.35% | 240 |
| 0.50 | 0.39 | 56.74% | **68.17%** | 94.42% | 378 |

(d) Ablation on the number of blocks. 'a-b-c' is the number of 1-2-3D blocks, correspondingly.

| #Blocks | DD ↑ | IQ ↑ | SC ↑ | Time ↓ |
|---|---|---|---|---|
| 8-24-36 | 57.25% | 66.57% | 93.59% | 172 |
| **8-48-72** | 56.91% | 67.45% | **94.72%** | 187 |
| 24-48-144 | **57.70%** | **68.60%** | 94.42% | 222 |

## 4.4 ABLATION STUDY

**Block partitioning strategy.** We ablate the layer-wise recurrent block partitioning strategy by removing one of the 1D, 2D, or 3D partitioning patterns and show the results in Table 3a. Removing any single partitioning patter generally leads to a drop in performance. For instance, The 2-3D strategy achieves a slightly higher score but at the cost of increased training time. This suggests that the 1D partition contributes to efficiency. This highlights the benefit of incorporating diverse partitioning schemes to capture different aspects of spatio-temporal locality.

**Block choice design.** We ablate the key components of our block selection mechanism: global block selection and threshold-based selection, detailed in Table 3b. The results show that our proposed "threshold + global" strategy achieves the best performance across all metrics. Removing either leads to a performance drop. This indicates that both adaptively determining the number of blocks via a threshold and selecting blocks from a global pool are crucial for VMoBA's effectiveness.

**The influence of threshold $\tau$.** We investigate the impact of the threshold $\tau$. Results are presented in Table 3c. We observe that a larger threshold generally leads to better performance. However, this comes at the cost of increased training time. This is intuitive, as a larger $\tau$ means selecting more blocks, thereby increasing computational load but also enhancing the model's capacity. A threshold of 0.25 offers a good balance.

**The influence of block number.** We evaluate the influence of the number of blocks, as shown in Table 3d. Generally, using more blocks tends to bring better performance. However, this also increases the training time. Our default setting of "8-48-72" achieves comparable scores with a lower training cost, which provides a good trade-off.

## 5 CONCLUSION

This paper introduces VMoBA, a mixture-of-block sparse attention mechanism designed to address the computational challenges of training long-sequence data in VDMs. Our approach was motivated by a task-specific analysis of attention mechanisms in pre-trained video DiTs. VMoBA incorporates three key innovations: layer-wise recurrent block partitioning, global block selection, and threshold-based block selection. Extensive experimental results demonstrate VMoBA's effectiveness. It achieves speedups of up to 1.48x while maintaining or even improving generation quality compared to full attention. Moreover, VMoBA provides competitive speedups in training-free inference settings, showcasing its versatility.

**Acknowledgement.** This work is supported by the National Key Research and Development Program of China (No. 2023YFC3807600).

## ETHICS STATEMENT

Our research is grounded in ethical practices, with particular attention paid to the responsible use of data. All datasets employed in this study are publicly available and well-established within the computer vision community. Specifically, our training data includes Koala-36M (Wang et al., 2024b), while our benchmarking was conducted on VBench (Huang et al., 2024). Our use of this data is in accordance with their provided licenses and intended academic purpose.

## REPRODUCIBILITY STATEMENT

We are committed to ensuring the reproducibility of the research presented in this paper. To this end, comprehensive implementation details for our models and experiments are provided in Appendix, including the training procedures and all hyperparameters used. Furthermore, upon acceptance of this paper, all source code, datasets, and trained model checkpoints will be made publicly available.

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

# Appendix

**Table of contents.** The supplementary includes the following sections:

## A  IMPLEMENTATION DETAILS

Table 4: **Implementation detials** of experiments in the main paper.

| Video Resolution | Training-free | | Training | |
|---|---|---|---|---|
| | $81 \times 480 \times 832$ | $81 \times 720 \times 1280$ | $93 \times 576 \times 1024$ | $141 \times 480 \times 832$ |
| Latent Resolution | $21 \times 30 \times 52$ | $21 \times 45 \times 80$ | $24 \times 36 \times 64$ | $36 \times 30 \times 52$ |
| Temporal Block Size | 3 | 3 | 3 | 3 |
| Spatial Block Size | $5 \times 13$ | $9 \times 10$ | $6 \times 8$ | $5 \times 13$ |
| Spatial-Temporal Block Size | $7 \times 5 \times 13$ | $7 \times 15 \times 20$ | $8 \times 12 \times 8$ | $12 \times 10 \times 13$ |
| Temporal Blocks | 7 | 7 | 8 | 7 |
| Spatial Blocks | 24 | 40 | 48 | 24 |
| Spatial-Temporal Blocks | 72 | 36 | 72 | 36 |
| Block Selection Type | Local + TopK | Local + TopK | Global + Threshold | Global + Threshold |
| $k$ | 2 ǀ 6 ǀ 18 | 2 ǀ 10 ǀ 9 | - | - |
| $\tau$ | - | - | 0.25 | 0.25 |

The detailed hyperparameter settings of the main experiments are shown in Table 4. Note that apart from the 3D block partition algorithm, we use local and topk query-key block selection in the training-free experiments, which is the same option as the original MoBA (Lu et al., 2025). We empirically find that adopting VMoBA's block selection strategy will cause vibration effects on the generated videos. On the contrary, the original local and topk options can better preserve the generation stability, probably due to their closer modeling nature to full attention. However, we use the full VMoBA setting in all training experiments, observing a stronger performance and efficiency gain compared to MoBA, which demonstrates the effectiveness of VMoBA as a sparse attention algorithm designed for training video diffusion models.

## B  ADDITIONAL EXPERIMENTS

Table 5: **Training results** on original and shorter sequence lengths.

| Video Size | Method | Sparsity | Quality | | | | | Efficiency | |
|---|---|---|---|---|---|---|---|---|---|
| | | | TextConsis ↑ | Dynamic ↑ | BGConsis ↑ | ImageQual ↑ | SubConsist ↑ | FLOPs ↓ | Training Time ↓ |
| | Pretrained Wan 2.1 | - | 27.85% | 73.45% | 94.47% | 62.06% | 90.70% | 282.64T | - |
| Origin | FullAttn | - | 20.32% | 98.43% | 92.39% | 61.18% | 85.09% | 282.64T (1.00×) | 104 GPU Hours (1.00×) |
| $81 \times 480 \times 832$ | + DiTFastAttn (Yuan et al., 2024) | 0.50 | 20.95% | 95.87% | 85.08% | 58.86% | 81.22% | 182.73T (1.55×) | - |
| | + SVG (Xi et al., 2025) | 0.50 | 19.11% | 97.96% | 93.06% | 60.13% | 84.12% | 182.92T (1.54×) | - |
| 33K Tokens | MoBA (Lu et al., 2025) | 0.25 | 22.15% | 5.40% | 98.27% | 64.78% | 97.75% | 133.50T (2.12x) | 126 GPU Hours (0.82×) |
| | VMoBA | **0.19** | 21.72% | 69.35% | 97.16% | **65.44%** | **97.92%** | 149.00T (1.90x) | 104 GPU Hours (1.00×) |
| | Pretrained Wan 2.1 | - | 27.13% | 86.31% | 94.17% | 63.60% | 91.49% | 67.73T | - |
| Shorter | FullAttn | - | 22.13% | 49.10% | 98.15% | 62.84% | 96.42% | 67.73T (1.00×) | 88 GPU Hours (1.00×) |
| $81 \times 320 \times 512$ | + DiTFastAttn (Yuan et al., 2024) | 0.50 | 22.16% | 48.37% | 94.46% | 62.23% | 91.00% | 51.09T (1.32×) | - |
| | + SVG (Xi et al., 2025) | 0.50 | 20.99% | 48.70% | 97.84% | 62.03% | 95.70% | 51.17T (1.32×) | - |
| 13K Tokens | MoBA (Lu et al., 2025) | 0.25 | 24.05% | 50.96% | 98.08% | 62.86% | 97.70% | 42.84T (1.58×) | 118 GPU Hours (0.75×) |
| | VMoBA | **0.18** | **24.29%** | 38.40% | 98.09% | **62.92%** | **98.78%** | **40.48T (1.67×)** | 103 GPU Hours (0.86×) |

We conduct additional training experiments on the original and shorter sequence lengths and show the results in Table 5. For the original sequence length (33K), VMoBA's speed is very close to the pre-trained full attention model, while for a shorter sequence length (13K), VMoBA becomes even slower. This is reasonable because the acceleration of VMoBA is more significant when the sequence length goes longer, as shown in Fig. 1. Moreover, in these two experiments, VMoBA consistently outperforms MoBA in both performance and efficiency, further demonstrating its effectiveness.

## C    PRE-TRAINING LOSS COMPARISON

Table 6: **Model configuration** of the pre-training experiments.

| Parameter | Layer | Head | Hidden Size | FFN Dim |
|---|---|---|---|---|
| 5.6M | 3 | 3 | 384 | 896 |
| 60.4M | 9 | 5 | 640 | 2688 |
| 217M | 15 | 7 | 896 | 4480 |
| 526M | 21 | 9 | 1152 | 6272 |

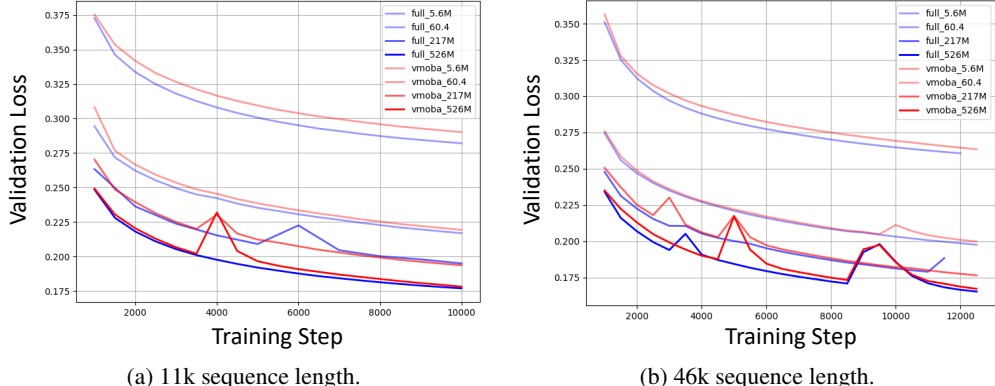

(a) 11k sequence length.

(b) 46k sequence length.

Figure 7: **Validation loss comparison for pre-training** between VMoBA and full attention. Blue and red lines are full attention and VMoBA, respectively. Deeper colors represent bigger model sizes.

We conduct training-from-scratch experiments to evaluate the pre-training ability of VMoBA, compared with full attention. Following MoBA (Lu et al., 2025), we train on two sequence lengths, 11k ($77 \times 288 \times 512$) for shorter length, and 46k ($77 \times 576 \times 1024$) for longer length. The 1-2-3D block numbers of VMoBA are 10, 48, and 60. The threshold $\tau$ is set to 0.25. The model configuration are shown in Table 6. We use the same architecture as Wan 2.1 (Wang et al., 2025). For VMoBA models, we replace all the self-attention layers with VMoBA, which is the only change. The results for 11K length are shown in Fig. 7a. Despite VMoBA's validation loss curves being consistently higher than those of full attention, the difference between them is shrinking as the model size increases. This indicates that VMoBA performs closely with full attention for larger model scales. Moreover, the results for 46K length are shown in Fig. 7b. We observe close overlapped validation loss curves between VMoBA and full attention pre-training, further demonstrating that VMoBA potentially performs better at longer sequences, which is also revealed by MoBA (Lu et al., 2025). In conclusion, the pre-training experiments demonstrate that VMoBA is a reliable alternative to full attention for the pre-training of video generation models.

## D    LIMITATIONS AND FUTURE WORK

As shown in Table 5, the speed up of VMoBA is sub-optimal when the sequence length is short, even though its FLOPs are lower. This may be due to the inconsistent memory distribution of selected blocks. The current FlashAttention (Dao et al., 2022)-based implementation may not be able to optimize the speed of the sparse attention mechanism. Future work may focus on more efficient and hardware-friendly implementations to make the practical speed of VMoBA closer to the theoretical FLOPs.

## E    USE OF LLMS

In preparing this paper, LLMs are utilized as a general-purpose assistive tool. Specifically, the use of LLMs is strictly limited to proofreading the author-written text for grammatical errors, spelling corrections, and improvements to language clarity. This application is consistent with the use of

conventional grammar-checking software and did *not* extend to research ideation, data analysis, or the generation of any substantive content.

