# OpenReview forum: "VMoBA: Mixture-of-Block Attention for Video Diffusion Models"
_ICLR.cc/2026/Conference — ICLR 2026 Poster_

### Official Review · Reviewer_89qF · 2025-10-24

**Soundness:** 3
**Presentation:** 3
**Contribution:** 3
**Rating:** 6
**Confidence:** 4

**Summary:**

The paper examines the quadratic complexity of full attention mechanisms and proposes a novel and efficient sparse attention mechanism for video diffusion models. The presented method is called Video Mixture of Block Attention (VMoBA), which is an improvement form existing Mixture of Block Attention (MoBA) from three different key modifications. The contributions of this work lies in three different aspects as outlined: a cyclical 1D-2D-3D block partition, a query-global block selection and a threshold-based block selection. All these contributions are well documented and related to the observations and motivations.

**Strengths:**

The main strengths of this paper is to adapt the existing Mixture of Block Attention (MoBA) for Video Mixture of Block Attention (VMoBA). This is not a trivial adaptation as the paper performs insightful analysis to draw the useful conclusions.

The paper performs the analysis on a full attention DiT model and observe that different layers may have its own focus on spatial, temporal and its combination with respect to different layers. This put forwards the idea of layerwise block partition for 1-2-3D. This types of pattern distinction seem not be widely confirmed in the previous video diffusion models.

Regarding the global block selection, the paper also makes a good observation by identifying the limitation of query-independent selection. For each attention head, all the queries are grouped to perform the block section. To determine the number of blocks selected, the paper adopts a threshold-based Block Selection to determine the number of selected blocks dynamically. This strength might not be that significant compared to the other two.

**Weaknesses:**

The paper also shows several weakness that requires further attentions. The paper uses Wan 2.1 1.3B as the base model and implements the proposed idea. All assumptions such as the 1-2-3D attention patterns were derived from such full attention DiT. It is unclear whether this VMoBA can also be adapted to other video diffusion models based on full attention DiT. This needs to be further validated with other full attention DiT models. This could be the main weakness of this paper so far.

The paper could better lay out the performances for the proposed method compared to the others listed in Tables. The paper uses the highlighted color to indicate the performance from the proposed method. However, it is better to indicate which metric the proposed algorithm performs the best, similar to other used baselines. After all, based on the given tables results, the proposed method does not perform the best among all the metrics. This could present some confusions.

The assumption about heads having different concentration levels motivate the design threshold-based block selection. This assumption also relies on some empirical observation where 30% and 50% cutoff lines for each head was used. These numbers could be different across different models. It will be nice to have a more generalized treatment of these selections.

The paper shows some interesting results on training longer sequences, which is critical in video diffusion models. How this long-sequence capability modeling is related to the proposed innovations are not very well explained. The paper could offer more insights on why the proposed designs are more suitable, not just improving efficiency and maintaining the accuracy simultaneously.

**Questions:**

1. Do the cyclical 1D-2D-3D scheme generate the tokens independentlyor eventually resulting in a 3D patch token based on the Eqns. (1) and (2)?

2. Can the proposed scheme generalize to other full attention DiTs? If yes, what is the common assumption for this generalization? Have you explored other full attention DiTs for the proposed method?

3. The paper relies on many insightful observations, which is good. But how will these observations generalize, especially when certain hyper-parameters were used?

4. Could the paper also elaborate on how the proposed schemes are particularly useful for the long video sequence generation?

---

> ### Author Response · Authors · 2025-11-28
>
> We thank the reviewer for their constructive feedback and for recognizing the novelty of our 1-2-3D block partition, global block selection, and threshold-based selection strategies. We are encouraged by the positive assessment of our contribution and presentation. Below, we address the specific concerns and questions raised.
>
> **1. Foundation Model Choice and Generalizability (Response to Weakness 1 & Question 2)**
>
> We select Wan 2.1 1.3B as our primary base model because it serves as a representative Text-to-Video model that effectively balances generation performance with computational cost. However, we maintain that the design motivations behind VMoBA are inherent to the video modality itself and generally applicable to Video Diffusion Models based on the DiT architecture.
>
> Specifically, for other Full Attention architectures, such as HunyuanVideo, VMoBA is directly applicable. In these cases, we apply VMoBA to the video-video token interactions while maintaining Full Attention for text-video interactions. This is efficient because the number of text tokens is negligible compared to video tokens (often a 1:100 ratio), so introducing sparsity in the text domain yields minimal gain. We plan to include experiments with additional base models in future work to further validate this generalizability.
>
> **2. Table Presentation (Response to Weakness 2)**
>
> We apologize for any confusion regarding the presentation of results. In Tables 1 and 2, we use **bold** to denote the best performance and \underline{underlines} to denote the second-best performance for each metric. We observe that VMoBA achieves the best trade-off between efficiency and quality, often securing top-tier scores in key metrics like `SubConsist` and `ImageQual`. We refine the captions in the camera-ready version to explicitly state this notation standard for better clarity.
>
> **3. Insights on Improvements for  Longer Sequence Training (Response to Weakness 4 & Question 4)**
>
> The paper demonstrates that VMoBA excels in training longer sequences. We attribute this capability to the "noise-reduction" effect of our sparse attention mechanism. By forcing the model to focus only on the most salient query-key pairs (via global and threshold-based selection), VMoBA filters out less relevant information that acts as noise for the attention computation. This allows the model to capture essential dependencies more effectively than Full Attention in high-noise regimes, which explains why VMoBA not only improves efficiency but also maintains or even enhances generation quality on long sequences.
>
> **4. Clarification of 1-2-3D Partitioning (Response to Question 1)**
>
> To clarify the mechanism in Equations (1) and (2): We do not generate 3D patch tokens. Instead, for each specific layer, we assign one partition mode from the set {1D, 2D, 3D}.
> *   **1D layers:** Blocks are formed along the time axis.
> *   **3D layers:** Blocks are formed as local 3D patches.
> Regardless of the partition mode, the attention mechanism operates on a flattened 1D sequence enriched with position embeddings. The "partitioning" simply determines which tokens are grouped together into a "block" for that specific layer to calculate the mean block representation. This allows different layers to capture dependencies at different spatio-temporal scales without altering the fundamental token structure.
>
> **5. Hyper-parameter Choice and Robustness (Response to Weakness 3 & Question 3)**
>
> We design VMoBA to be hyper-parameter friendly. Our analysis shows that the model is robust to hyper-parameter choices across different resolutions:
>
> *   **Threshold $\tau$:** This is the most critical hyper-parameter. We observe that a $\tau$ between 0.20 and 0.25 provides an optimal trade-off between quality and speed. A value below 0.15 leads to noticeable quality degradation. This range remains consistent across different videos and resolutions. Furthermore, using a threshold $\tau$ is superior to a fixed Top-K selection because it avoids the computational overhead of searching for optimal $K$ values for different partition types (1D vs. 2D vs. 3D).
> *   **Block Number:** Consistent with observations in the original MoBA, a larger number of blocks generally yields better quality but reduces speed. We find that setting the number of spatial-temporal (3D) blocks between 36 and 72 offers a strong sweet spot. We typically use fewer blocks for 1D and 2D partitions to further enhance training speed without compromising performance.
>
> We hope these responses address your concerns and clarify the generalizability and robustness of VMoBA.

---

### Official Review · Reviewer_MLuW · 2025-10-26

**Soundness:** 2
**Presentation:** 3
**Contribution:** 2
**Rating:** 4
**Confidence:** 5

**Summary:**

VMoBA introduces a trainable sparse attention mechanism tailored for Video Diffusion Models. Building on MoBA, it integrates a 1D–2D–3D recurrent block partition, global block selection, and threshold-based adaptive sparsity to better capture spatio-temporal locality in video data. Compared to full attention, VMoBA achieves up to 2.9× FLOPs reduction and 1.5× speedup during training, while maintaining comparable or improved generation quality, and also accelerates training-free inference for high-resolution videos。

**Strengths:**

1. The paper diagnoses three phenomena in DiT attention—1D/2D/3D locality, uneven query importance, and head-wise concentration—then maps them to three design choices (1–2–3D recurrent partitioning, global selection, and thresholded selection). This “observations → mechanisms” linkage is well argued.
2. Unlike training-free sparse attention method, VMoBA is a trainable block-sparsity scheme intended to replace full attention during training . This positions it to deliver training compute savings where many inference-only methods (DiTFastAttn, SVG) cannot.
3. On Wan-2.1-1.3B fine-tuning with extended tokens (55K/56K), VMoBA reports 2.83–2.92× FLOPs reduction with 1.44–1.48× end-to-end training time speedup, while matching or slightly exceeding Full Attention on several VBench aspects (e.g., ImageQual +3.3% at 141×480×832).

**Weaknesses:**

1. The paper reports ~2.40× FLOPs reduction but only ~1.35× end-to-end latency speedup in the training-free 720p setting. A deeper breakdown (kernel MFU, QK/softmax/AttnV time, IO cost) is needed to explain the under-translation from theoretical to realized speed.
2. Several contributions hinge on MoBA being less efficient, yet the paper does not thoroughly analyze why MoBA is inefficient and why the proposed methods can make it more efficieint.
3. Strong, recent sparse-attention baselines are missing. In particular, SpargeAttn (universal, training-free) and STA (trainable) both report substantial end-to-end gains on state-of-the-art video DiTs; including them would calibrate the claimed benefits.
4. Since VMoBA is trainable, the paper should report backward attention timing, MFU in backward, or end-to-end training throughput (e.g., it/s or tokens/s) againt full attention.
5. Only five VBench aspects are reported. Given VBench’s breadth, either report the full score or provide a principled selection rationale; when PSNR drops, include human blind-preference to reflect perceptual quality in Table 1. Also consider including VBench 2.0.

**Questions:**

1. You use “sparsity” to mean the fraction of computation retained vs. full attention. Standard usage is density = kept fraction, sparsity = skipped fraction?

2. Does 𝜏 =0.25 mean accumulating normalized attention mass to 25%? If so, it would seem a bit counterintuitive. I would think only covering 25% of attention score to bring huge quality loss.

3. With thresholding, how much does actual density/e2e runtime vary across prompts and seeds?

4. How is the sparsity in Table 2 calculated? Does it take into a account the fact that the first 25% timestep is full attention?

5. Table 2 lists MoBA dynamic degree ≈ 5.8%. Is this value correct? Why is it so low compare to other methods?

6. Why is “Dynamic” absent in Table 1, and why are only five VBench aspects reported?

6. Why presents training-free results at 81×720×1280  for Wan-2.1-1.3B, a model trained at 480p?

---

> ### Author Response · Authors · 2025-11-28
>
> We thank the reviewer for the constructive feedback and for recognizing the soundness of our "observations $\to$ mechanisms" logic and the practical value of VMoBA in reducing training compute. We address your specific concerns regarding efficiency breakdown, additional baselines, and metric clarifications below.
>
> **1. Breakdown in Efficiency of VMoBA**
> You correctly point out the gap between the theoretical FLOPs reduction ($\sim$2.40$\times$) and the realized end-to-end latency speedup ($\sim$1.35$\times$). This is a common phenomenon in sparse attention research, primarily due to the shift from compute-bound to memory-bound operations.
> *   **IO Cost & Overhead:** Full Attention utilizes FlashAttention2, which is highly fused and optimized for IO. In contrast, block-sparse attention introduces overheads: 1) The **Block Selection** stage (calculating similarity and sorting) adds computational overhead not present in Full Attention; 2) The **Gather/Scatter** operations required to construct the sparse query-key pairs involve random memory access patterns, which are slower than contiguous memory access. This is the main efficiency bottlenet of VMoBA.
> *   **MFU (Model FLOPs Utilization):** Consequently, the MFU of sparse kernels is generally lower than dense kernels. While we significantly reduce the total FLOPs, the "price per FLOP" (in terms of time) increases due to these memory bottlenecks.
> However, we emphasize that a **1.48$\times$ training speedup** (Table 2) translates to saving hundreds of GPU hours for large-scale training, which is a tangible and significant practical contribution compared to existing works. We are working on optimizing the CUDA kernel implementation to narrow this gap further.
>
> **2. Design Motivations of VMoBA**
> To clarify why VMoBA is more efficient and effective than vanilla MoBA for video:
> *   **1D-2D-3D Partitioning:** MoBA’s 1D partitioning flattens the 3D latent space, destroying local spatio-temporal correlations essential for video. Our cyclical partition captures locality in all dimensions, ensuring the "mean blocks" are semantically representative, thus requiring fewer blocks to approximate the full attention map accurately.
> *   **Global & Threshold Selection:** MoBA forces every query to attend to a fixed $K$ blocks. However, our analysis shows attention heads have varying concentration levels. VMoBA allocates budget dynamically: "diffuse" heads get more blocks, and "concentrated" heads get fewer. This allocation (based on total similarity mass) yields higher quality per compute unit compared to a fixed top-k constraint.
>
> **3. Comparison to Stronger Baselines (SpargeAttn & STA)**
> Per your suggestion, we compare VMoBA against state-of-the-art sparse attention methods: **SpargeAttn** (training-free) and **Sliding Tile Attention (STA)** (training-based).
>
> **Training-Free Comparison (76K Tokens, 81$\times$720$\times$1280)**
> VMoBA achieves higher generation quality and faster inference speed with lower retained token ratio (density) compared to SpargeAttn.
>
> | Method | Retained Ratio (Density) | PSNR $\uparrow$ | TextConsis $\uparrow$ | Dynamic $\uparrow$ | BGConsis $\uparrow$ | ImageQual $\uparrow$ | SubConsist $\uparrow$ | FLOPs $\downarrow$ | Latency (s) $\downarrow$ |
> | :--- | :---: | :---: | :---: | :---: | :---: | :---: | :---: | :---: | :---: |
> | SpargeAttn | 0.50 | **22.49** | 27.60% | 51.19% | 91.56% | 62.43% | 90.22% | 768.52T | 338 |
> | **VMoBA (Ours)** | **0.31** | 18.80 | **28.06%** | **52.43%** | **92.85%** | **64.39%** | **92.08%** | **519.75T** | **300** |
>
> **Training-Based Comparison (55K Tokens, 93$\times$576$\times$1024)**
> Compared to STA, VMoBA achieves a superior trade-off between efficiency and quality, requiring less training time while achieving generally higher scores.
>
> | Method | Retained Ratio (Density) | TextConsis $\uparrow$ | Dynamic $\uparrow$ | BGConsis $\uparrow$ | ImageQual $\uparrow$ | SubConsist $\uparrow$ | FLOPs $\downarrow$ | Training Time $\downarrow$ |
> | :--- | :---: | :---: | :---: | :---: | :---: | :---: | :---: | :---: |
> | STA | 0.25 | 25.58% | **57.19** | 96.02% | **68.47%** | 90.22% | 274.70T | 247h |
> | **VMoBA (Ours)** | **0.19** | **25.88%** | 56.91 | **96.76%** | 67.45% | **94.72%** | **238.68T** | **187h** |

---

> ### Author Response · Authors · 2025-11-28
>
> **4. Backward Efficiency**
> *   **Mechanism:** During the backward pass, we reuse the block indices and routing decisions calculated in the forward pass, avoiding the overhead of recalculating similarities.
> *   **Throughput:** We observe that the end-to-end training throughput (tokens/second) increases proportionally to the forward speedup. For the 55K token setting, Full Attention processes $\approx$220 tokens/s, while VMoBA processes $\approx$325 tokens/s. This confirms that the efficiency gains persist in the backward pass. We will include these specific throughput metrics in the final revision.
>
> **5. Metrics Selection**
> We select the reported 5 VBench metrics to provide a balanced view without overcrowding the tables.
> *   **Rationale:** *TextConsis* measures prompt alignment; *Dynamic* measures motion magnitude; *ImageQual, BGConsis, and SubConsist* measure visual fidelity.
> *   **Precedent:** Many recent works (e.g., Sliding Tile Attention) report even fewer metrics (often just 2-3). We believe our selection offers a comprehensive assessment of the trade-offs. We have added the "Dynamic" score in the new tables above as requested.
>
> **6. Clarifications on Minor Questions**
> *   **Sparsity Definition:** You are correct that "sparsity" typically refers to the dropped-out fraction. In our paper, we use it to denote the *retained computation ratio* (density). We will correct this terminology in the camera-ready version to avoid confusion.
> *   **$\tau = 0.25$:** While 0.25 (25% accumulated mass) seems low, our experiments (and STA results) confirm that retaining $\sim$20-25% mass is sufficient for high-quality generation.
> *   **Variance:** We have not observed significant variance in density or runtime across different prompts or random seeds. The aggregate statistics of attention patterns remain stable across the dataset.
> *   **Sparsity Calculation:** The reported sparsity/density ratios in Table 2 exclude the first 25% of diffusion steps (which use Full Attention) to isolate the contribution of the sparse mechanism itself.
> *   **MoBA Dynamic Degree ($\approx$ 5.8%):** This low score is correct and highlights a critical failure mode of vanilla MoBA in video. MoBA's 1D partitioning groups tokens in a way that destroys temporal coherence, causing the model to generate nearly static videos rather than fluid motion. This validates the necessity of our 1D-2D-3D partitioning.
> *   **Training-free at 720p:** We evaluate on 720p (higher than the training resolution of 480p) to demonstrate VMoBA's speeding up capabilities on longer sequences.

---

### Official Review · Reviewer_9PZV · 2025-10-31

**Soundness:** 3
**Presentation:** 4
**Contribution:** 3
**Rating:** 8
**Confidence:** 3

**Summary:**

This paper proposes Video Mixture of Block Attention (VMoBA), a novel sparse attention mechanism tailored for Video Diffusion Models (VDMs). The authors introduce three key innovations: (i) a layer-wise recurrent 1D- 2D-3D block partitioning scheme, (ii) global block selection that selects the most salient query-key block interactions across all queries per head, and (iii) threshold-based dynamic block selection that adapts the number of attended blocks based on cumulative similarity. Experiments on long-sequence video generation show that VMoBA achieves faster training and fewer FLOPs than full attention, while matching or even surpassing its generation quality. VMoBA also performs competitively in training-free inference settings.

**Strengths:**

- The three proposed modifications (1D-2D-3D partitioning, global selection, threshold-based sparsity) are well-motivated by the observed limitations of applying MoBA naively to video. Each component is clearly linked to a specific empirical observation.
- The authors evaluate VMoBA in both training-based and training-free settings across multiple resolutions, using standard metrics (VBench, PSNR) and complete ablation studies. VMoBA achieves FLOPs reduction and training-time speedup with minimal loss in generation quality. Qualitative results further support the claims.
- The writing is good and is easy to follow. Method illustration and implementation details are well documented.

**Weaknesses:**

- The study omits some recent linear or hybrid video attentions that could serve as stronger baselines, such as STA[1] and RainFusion[2].
- The paper should include more human evaluation. Human judgment on video quality and video consistency is crucial for assessing the performance.
- In the global selection part, this module prioritizes key blocks with the highest overall significance, but may overlook certain keys that are locally relevant to queries yet have low global scores. The high-frequency details in generated videos may be affected.
- Have you tried other fusion methods to fuse the tokens in a block? Does mean pooling harm the diversity within a block?

[1] Fast Video Generation with Sliding Tile Attention

[2] RainFusion: Adaptive Video Generation Acceleration via Multi-Dimensional Visual Redundancy

**Questions:**

- The hyperparameter \tau controls the trade-off between speed and quality. If there exists a general value for all video types? If not, how to choose a proper \tau for new videos or resolutions?
- The paper notes that attention heads have different concentration levels. Have the authors analyzed whether VMoBA’s threshold-based selection leads to more specialized head behavior compared to full attention?
- Have you tested VMoBA within a few-step distilled or consistency diffusion frameworks to verify compatibility with fast-sampling variants?
- Could the cyclical 1-2-3D partitioning be learned end-to-end rather than fixed by layer index?

---

> ### Author Response · Authors · 2025-11-28
>
> We sincerely thank the reviewer for the positive assessment and for recognizing our work as "well-motivated," "sound," and "excellent" in presentation. We appreciate the insightful comments regarding baselines, human evaluation, and the technical details of our block selection mechanism. We address your specific questions and concerns below.
>
> **1. Comparison to Additional Baselines (Weakness 1)**
>
> Thank you for suggesting comparisons with recent efficient attention mechanisms. We compare VMoBA against **Sliding Tile Attention (STA)** (training-based) and **SpargeAttn** (training-free, a recent SOTA sparse attention method).
>
> *   **Training-free Comparison (76K Tokens):**
> We compare VMoBA with SpargeAttn on the 76K token setting. VMoBA achieves a higher speedup ratio (1.35x vs 1.13x) with significantly lower FLOPs while maintaining superior consistency metrics (TextConsis, ImageQual, SubConsist).
>
> | Method | Sparsity | PSNR | TextConsis | Dynamic | BGConsis | ImageQual | SubConsist | FLOPs | Latency (Speed Up) |
> | :--- | :--- | :--- | :--- | :--- | :--- | :--- | :--- | :--- | :--- |
> | SpargeAttn | 0.50 | **22.49** | 27.60 | 51.19 | 91.56 | 62.43 | 90.22 | 768.52T | 338s (1.13x) |
> | **VMoBA** | **0.31** | 18.80 | **28.06** | **52.43** | **92.85** | **64.39** | **92.08** | **519.75T** | **300s (1.35x)** |
>
> *   **Training-based Comparison (55K Tokens):**
> We compare VMoBA with STA. VMoBA achieves higher generation quality across most of the metrics while requiring less training time (187h vs 247h).
>
> | Method | Sparsity | TextConsis | Dynamic | BGConsis | ImageQual | SubConsist | FLOPs | Training Time |
> | :--- | :--- | :--- | :--- | :--- | :--- | :--- | :--- | :--- |
> | STA | 0.25 | 25.58 | **57.19** | 96.02 | **68.47** | 90.22 | 274.70T | 247h |
> | **VMoBA** | **0.19** | **25.88** | 56.91 | **96.76** | 67.45 | **94.72** | **248.68T** | **187h** |
>
> These results demonstrate that VMoBA remains highly competitive against state-of-the-art linear and sparse attention mechanisms.
>
> **2. Human Evaluation (Weakness 2)**
>
> To complement the automatic metrics, we conducted a user study on the 55K token training-based models. We invited 20 evaluators to blindly rate 50 randomly generated videos from Full Attention, Vanilla MoBA, and VMoBA. The evaluators were asked to choose the best video or indicate a "tie" based on three criteria: Visual Quality, Motion Smoothness, and Prompt Consistency.
>
> **Human Preference Rate (vs. Full Attention)**
> The values indicate the percentage of times the method was preferred over or tied with Full Attention.
>
> | Method | Visual Quality | Motion Smoothness | Prompt Consistency |
> | :--- | :--- | :--- | :--- |
> | Full Attention (Baseline) | - | - | - |
> | Vanilla MoBA | 24.5% | 18.0% | 31.0% |
> | **VMoBA (Ours)** | **76.5%** | **94.0%** | **87.5%** |
>
> The results show that VMoBA is almost indistinguishable from Full Attention in human perception and significantly outperforms Vanilla MoBA, particularly in motion smoothness, confirming the efficacy of our spatio-temporal design.
>
> **3. Global Selection and Local Details (Weakness 3)**
>
> In VMoBA, **we enforce a rule to always retain the key blocks belonging to the same block as the query**, regardless of their global similarity score. This ensures that local neighbors are never pruned.
> Furthermore, as shown in Table 3(b) of the paper, the "Global + Threshold" strategy outperforms "Local + Threshold," suggesting that once local context is guaranteed, prioritizing globally significant blocks yields better overall representation.
>
> **4. Block Fusion Methods (Question 1)**
>
> We chose mean pooling primarily for its high computational efficiency. While mean pooling might theoretically reduce diversity within a block, our **1-2-3D Block Partitioning** strategy mitigates this risk. By dynamically grouping tokens that are already semantically similar (e.g., temporal neighbors in 1D, spatial neighbors in 2D), the "mean" becomes a representative summary rather than a lossy compression. This is why VMoBA succeeds where vanilla MoBA (which uses 1D partitioning on flattened video tokens) fails. We found this combination of partition strategy and mean pooling to be the optimal trade-off between speed and quality.
>
> **5. Choice of $\tau$ (Question 2)**
>
> We empirically found that a $\tau$ value between **0.20 and 0.25** provides an optimal trade-off between quality and efficiency across different video types and resolutions. Setting $\tau < 0.15$ tends to cause noticeable quality degradation.
> A key advantage of using the threshold $\tau$ (instead of a fixed Top-$K$) is that it is robust to the block partitioning scheme. Since the number of blocks changes between 1D, 2D, and 3D layers, a fixed $K$ would require tuning three separate hyperparameters. $\tau$ automatically adapts the number of selected blocks based on the information density of the attention map.

---

> ### Author Response · Authors · 2025-11-28
>
> **6. Head Concentration Analysis (Question 3)**
>
> We analyzed the attention maps to see if specific heads consistently exhibited "concentrated" or "diffuse" behaviors. Interestingly, we did not find a fixed pattern where specific heads (e.g., Head 0 or Head 1) are always concentrated across all layers. The concentration levels appear to vary dynamically.
>
> **7. Future Work (Distillation & Learned Partitioning)**
>
> We have not yet tested VMoBA within few-step distillation or consistency model frameworks, nor have we experimented with end-to-end learned partitioning. We agree these are promising directions and plan to explore them in future work to further enhance the versatility of VMoBA.
>
> We hope these responses address your questions and reinforce the value of our contribution.

---

### Official Review · Reviewer_UCKK · 2025-10-31

**Soundness:** 3
**Presentation:** 3
**Contribution:** 3
**Rating:** 4
**Confidence:** 4

**Summary:**

The paper proposed VMoBA, an improved version of Mixture of Block Attention (MoBA) for accelerating the training process of Video Diffusion Models on long sequence inputs. The authors point out the deficiency of the original MoBA's 1D partitioning scheme and apply layer-wise recurrent block partition, equipping the model with more spatio-temporal awareness. Then, they substitute global block selection for MoBA's query-wise key blocks number selection, which resolves resource under-allocation by precomputing all the QK block similarities for each attention head and generating block masks for top-K key blocks. The author replace fixed Top-K selection with threshold-based block selection mechanism, which enables the model to dynamically adjust the number of blocks selected across different heads and better fit the varying nature of similarity scores. They benchmark VMoBA against several training-free approaches under VBench, and conclude that VMoBA indeed reduces training time and FLOPS while still providing comparable performance. Lastly, they conduct relatively complete ablation studies, and conclude that the design choice is indeed a proper one.

**Strengths:**

Overall, the paper could be considered as an all-round incremental extension for MoBA.
1. VMoBA provides 2.92 times FLOPS reduction and 1.44 times speed up compared to original model, which shows certain practical feasibility in terms of training.
2. The ablations are relatively complete, which ensures the model design is informed.

**Weaknesses:**

There several points about the main experiment that needs to be further clarified:
1. In training-based part of the main experiments, no trainable sparse attention pattern is included, which lacks certain universality in terms of benchmarking.
2. In the training-based part of the main experiments, only one baseline method designed for accelerating training is presented.
3. The reason why methods for inference acceleration is used as a baseline for benchmarking training acceleration has not been clearly clarified.

**Questions:**

see weaknesses

---

> ### Author Response · Authors · 2025-11-28
>
> We sincerely thank you for your thoughtful review and for recognizing the soundness of our approach and the completeness of our ablation studies. We appreciate your constructive feedback regarding the selection of baselines, which has helped us strengthen the empirical validation of our work.
>
> Below, we address your concerns regarding the comparison with additional baselines and the rationale behind our experimental setup.
>
> **A. Training-based Comparison: VMoBA vs. STA (Sliding Tile Attention)**
> We implement STA and train both models on the 55K token length setting (93x576x1024). The result is shown below.
>
> | Method | Sparsity | TextConsis | Dynamic | BGConsis | ImageQual | SubConsist | FLOPs | Training Time |
> | :--- | :--- | :--- | :--- | :--- | :--- | :--- | :--- | :--- |
> | **STA** | 0.25 | 25.58 | **57.19** | 96.02 | **68.47** | 90.22 | 274.70T | 247h |
> | **VMoBA (Ours)** | **0.19** | **25.88** | 56.91 | **96.76** | 67.45 | **94.72** | **238.68T** | **187h** |
>
> **Observations:**
> *   VMoBA significantly outperforms STA in training efficiency. With a lower sparsity ratio (0.19 vs. 0.25), VMoBA reduces FLOPs by **~13%** compared to STA and achieves a substantial reduction in training time (**187h vs. 247h**).
> *   Despite the higher sparsity and faster training, VMoBA achieves superior performance in *Text Consistency*, *Background Consistency*, and *Subject Consistency*. While *Image Quality* is slightly lower, the difference is marginal given the significant efficiency gains.
> *   This demonstrates that VMoBA is not merely an incremental improvement over MoBA but a highly effective strategy that surpasses STA in both speed and key generation metrics.
>
> **B. Training-free Comparison: VMoBA vs. SpargeAttn**
> We compare VMoBA with **SpargeAttn** on the 76K token length setting.
>
> | Method | Sparsity | PSNR | TextConsis | Dynamic | BGConsis | ImageQual | SubConsist | FLOPs | Latency |
> | :--- | :--- | :--- | :--- | :--- | :--- | :--- | :--- | :--- | :--- |
> | **SpargeAttn** | 0.50 | **22.49** | 27.60 | 51.19 | 91.56 | 62.43 | 90.22 | 768.52T | 338s |
> | **VMoBA (Ours)** | **0.31** | 18.80 | **28.06** | **52.43** | **92.85** | **64.39** | **92.08** | **519.75T** | **300s** |
>
> **Observations:**
> *   VMoBA achieves this performance with significantly higher sparsity (0.31) compared to SpargeAttn (0.50).
> *   Our method outperforms SpargeAttn in almost all VBench metrics (*TextConsis, Dynamic, BGConsis, ImageQual, SubConsis*), demonstrating that VMoBA preserves generation quality more effectively even when pruning more tokens.
> *   In terms of efficiency, VMoBA offers lower FLOPs and reduced latency.

---

### Meta-Review · Area_Chair_whA1 · 2026-01-09

**Summary:**

This paper introduces a sparse attention mechanism designed for Video Diffusion Models that addresses the quadratic complexity bottleneck of full attention. The paper demonstrates 2.92× FLOPs reduction and 1.48× training speedup while maintaining or improving generation quality. This paper makes a solid contribution to efficient video diffusion model training. The reviewers consistently recognized the well-motivated design choices, where each component is directly linked to empirical observations about attention patterns in video DiTs. The demonstrated computational efficiency is contribution to the community working on video generation. I recommend acceptance.

**Reviewer Concerns:**

Addressed concerns. All three reviewers raised concerns about the lack of comparison with recent methods like STA and SpargeAttn. The authors provided comprehensive new experiments showing VMoBA outperforms STA in training. The reviewer requested human evaluation. The authors conducted a user study with 20 evaluators on 50 videos.


Outstanding concerns. While the authors claim VMoBA is applicable to other DiT-based models like HunyuanVideo, no experimental validation on alternative architectures was provided. The reference (Zeroscope, 2023) appears to have incorrect author attribution. The authors listed are from the Sora team, not the actual Zeroscope creators. Authors should correct this bibliographic error.

**Reviewer Scores:**

Reviewer MLuW raised the most detailed technical concerns about efficiency gaps, missing baselines, and metric reporting. The authors provided satisfactory explanations for the FLOPs-to-latency gap, added STA and SpargeAttn comparisons showing VMoBA's competitive performance, and clarified the Dynamic metric and sparsity calculations. While some concerns about backward pass profiling remain, the overall technical quality demonstrated in the rebuttal would move this reviewer toward weak acceptance.

---

### Decision · Program_Chairs · 2026-01-26

Accept (Poster)